# The Extent and Impact of COVID-19 Infection among Family and Friends: A Scoping Review

Michael Wiblishauser *, Tori Chapa and K'Ondria Ellis

College of Education & Health Professions, University of Houston-Victoria, Victoria, TX 77901, USA; chapatp@uhv.edu (T.C.); elliskj@uhv.edu (K.E.)
* Correspondence: wiblishauserm@uhv.edu

**Abstract:** There exist a plethora of studies examining the psychological and physical impacts of COVID-19 on infected victims. Fewer studies have been published assessing the different types of impacts that an individual's COVID-19 infection has on close friends and family members. This is the first scoping review to gauge the reported psychosocial issues and daily hassles that impact the relatives and friends of infected individuals. This study was conducted by inputting key terms/MeSH terms into selected internet databases to locate prospective studies. The frameworks of scoping reviews by Arksey et al. and the preferred reporting items for systematic review and meta-analyses (PRISMA) were utilized in the methodology for identifying and selecting the studies. After data extraction, 37 studies were deemed suitable for analysis. The findings generated from each study were placed into combined categories. A total of 16 combined categories were generated from the amalgamation of the findings. The results show that psychosocial feelings (e.g., anxiety, stress, and depression) were the category with the highest prevalence of grouped findings. The results from this study may serve as the impetus for future interventions targeting the alleviation of psychosocial feelings or day-to-day hassles associated with having a loved one inflicted with a severe illness.

**Keywords:** COVID-19 victims; family members; friends; psychosocial impacts; emotional health; informal caregiving

## 1. Introduction Section

### 1.1. The Background

During the winter of 2019, the emerging SARS-CoV-2 virus (COVID-19) made both its sudden entry and impact felt throughout the world. The initial outbreak of the disease originated in China and soon spread throughout the world [1]. The World Health Organization (WHO) labeled the disease a pandemic due to its speed of transmission and its severity in causing health complications in at-risk populations [2]. As of late March 2023, there had been more than 762 million cases and almost 7 million deaths worldwide [3]. European countries reported the most confirmed cases, and African nations reported the fewest cases [3]. The pandemic led to societal lockdowns and social distancing/quarantining protocols, which adversely affected individuals in many countries. The lockdowns had not only social consequences but also economic ones. In the United States, it was projected that the pandemic would incur $16 billion in financial costs [4].

The impact of COVID-19 on infected victims may not only lead to physical effects but also emotional/mental ones. Individuals who both contacted COVID-19 and were successfully treated may suffer as much from the psychological vestiges of the disease as from the physical ones. Having been infected with the virus, a victim may suffer from a slew of post-exposure psychological symptoms (e.g., anxiety, depression, and stress) [5]. According to one estimate, almost one out of five COVID-19-infected individuals was at risk for developing at least one psychiatric disorder [6]. Even upon successful discharge, hospitalized individuals reported suffering from both social stigma and psychological

symptoms [7]. In one study, more than two-thirds of COVID-19 survivors reported suffering from symptoms associated with post-traumatic stress disorder (PTSD) [8]. PTSD symptoms may be reported with greater prevalence and severity when the survivors were recently discharged from hospital care vs. having been discharged for a longer period [9].

The pandemic contributed to an increased need for support for those infected with COVID-19. Strong family support was linked to an increase in the self-efficacy or confidence of COVID-19 patients to handle the daily stressors or hassles of life [10]. The lockdowns reportedly had a maladaptive impact on depression sufferers, who valued the importance of being present with family members or being able to be physically comforted by them [11]. Those patients who were hospitalized or placed in long-term care were particularly affected by the social distancing protocols [12], which could be linked to a lack of social and/or emotional support. Having a strong familial or social support system can be viewed as a strong protective factor against the psychosocial issues stemming from the pandemic. A trusting family support system not only assists in helping alleviate psychosocial issues but also reinforces more positive attitudes and/or increased acceptance of social distancing practices [13] and the increased adoption of personal protective equipment.

The pandemic has heralded new mediums for communication between healthcare providers and patients' family members. On many occasions, the family serves as informal caregivers for severely ill patients. Healthcare professionals need to be cognizant that many families of hospitalized COVID-19 victims may also feel constant pressure to manage their daily affairs. The hospitalization of their loved ones, coupled with day-to-day hassles, may create emotional strain for family members [12].

Since having a hospitalized family member can be stressful, effective communication to keep them abreast of the patient's status can be of paramount importance. It is recommended, even with physical distancing measures, that to assist in preserving family unity, family members be allowed limited physical presence near the bedsides of hospitalized loved ones [14]. If social distancing or other physical restrictions are placed on hospital visits, providing teleconferencing options or virtual reality visits may be the best options for solidifying communication between patients and their loved ones [14,15].

Due to the social distancing restrictions on hospital visits amidst highly communicable diseases, the use of technology is increasingly utilized as a mode of communication for relatives with their hospitalized loved ones or with healthcare providers. Teleconferencing [15], videoconferencing [16], or innovative family-support teams [17] can help hospitalized patients and healthcare providers effectively communicate with patients' relatives or close friends. Online psychosocial support groups can be employed as great assets in moderating the effects of COVID-19-linked caregiving stress [18]. It is deemed imperative that family members, especially those who are caring for sickly patients, be concerned for their own emotional health and physical well-being [19].

During the peak of the pandemic, many societal issues impacted the average person. Due to the societal lockdowns, many Americans suffered economic setbacks. In a survey by Karpman et al. [20], nearly 40% of respondents reported either losing employment, reduced work hours, or loss of work-related income resulting from the pandemic. Those from lower socioeconomic backgrounds may be more adversely affected. One survey study conducted during the peak of the pandemic found that 43% of family members worried about their employment, 69% worried about the availability of food, and 50% were concerned about the affordability of food products [21].

The lockdowns and their accompanying social distancing restrictions adversely affected the healthcare systems of many countries [22]. Many indigent individuals had difficulty accessing and acquiring needed services. Those who lack proper financial resources are more likely to be diagnosed with illnesses in their advanced stages, making it difficult to make full recoveries. The increased stress from financial or economic uncertainties, may lead to declines in physical and mental health [23]. One study found that during the pandemic, elderly individuals who lost their jobs and women were more likely to develop symptoms of depression compared to other groups [24].

*1.2. The Current Study*

The COVID-19 pandemic has profoundly affected how relatives and friends socialize and care for their diagnosed loved ones. The authors believe that the mental ramifications and daily hassles stemming from having an infected loved one may lead to declines in emotional, mental, and physical health. The pandemic has led to a greater focus on research assessing the mental health outcomes of COVID-19-infected individuals.

Currently, there is a dearth of research highlighting the reported mental or emotional health outcomes of the relatives and friends of COVID-19-infected individuals. To verify previous scoping reviews centered on the research question, the authors conducted literature searches through various databases (e.g., Google Scholar, PubMed, Medline) employing terms such as "scoping reviews", "COVID-19" and "friends or family members". These searches yielded no previously published scoping reviews.

The findings from this scoping review may optimistically lead to interventions targeting the alleviation of some issues that may affect relatives or close friends of a loved one either diagnosed with COVID-19 or other severe diseases. The findings may also facilitate future interventions in strengthening or stabilizing the relations between friends and family members, particularly those that may serve as informal caretakers or sources of social support. This scoping review's aim is to provide answers to the following research question: What are the reported psychosocial impacts and daily hassles among family members and friends of COVID-19-infected individuals?

## 2. Methods

The methodology for this scoping review was conducted using the first three steps of the framework devised by Arksey et al. [25]. This framework was selected due to its current usage and acceptance among professionals publishing scoping reviews. The first step is identifying and articulating the research question, which guided this scoping review. The authors collected and assessed published articles studying the psychosocial impacts and perceived daily hassles (e.g., economic or social hardships) of friends and family members of COVID-19-infected individuals.

The second step in the scoping review was identifying the relevant published studies to answer the research question. The review was conducted using the PubMed, Google Scholar, and Medline internet databases. Also employed were reference lists or bibliographies of studies found throughout the database searches. The database searches only included studies published in the English language. The authors employed the keywords/MeSH terms "COVID-19 patients", family members", and "friends" for the database searches. These keywords/MeSH terms were taken from the research question. Prospective articles were included if they contained the keywords in the titles or in the abstracts. Further inclusion of articles was conducted by reading and assessing the full-text articles.

The third step was selecting the articles to be part of the study's analysis. Only those articles that were published in scholarly, peer-reviewed journals and specifically covered COVID-19 were used in the scoping review. Articles covering other diseases were excluded. Studies that were published on websites, newspapers, or newsletters were excluded from the analysis. Articles were included that involved measuring the emotional health or the daily hassles of relatives and friends of COVID-19 diagnosed individuals. Duplicate articles (those articles that more than one researcher collected) were excluded from the review. Also, several studies were excluded due to inadequate coverage of the studied components in the review (e.g., lack of descriptive information on participants). Since the authors were looking to gauge findings associated with relatives and friends of COVID-19, studies that covered interventions were excluded from the scoping review.

## 3. Results Section

The initial databases' searches, utilizing the search terms, yielded 9202 results. After perusing through the searches, the researchers found 45 studies that met the inclusionary criteria for the review. After retrieval and thorough screening, eight more studies

were excluded. Lastly, a total of 37 studies were deemed appropriate for the scoping review [26–62] (Table 1). The flow of the studies' selection is exhibited through a modified preferred reporting items for systematic reviews and meta-analysis (PRISMA) [63] (Figure 1).

**Table 1.** Selected Studies from the Scoping Review.

| Author(s) | Participants' Relationship to Victim; Country | Type of Study | Main Findings |
|---|---|---|---|
| Berntzen et al., 2023 [26] | 12 family members of COVID-19-infected patients. (Norway) | Interview (Qualitative) | Caring for COVID-19 patients and fear of worsening self-health. Charged with responsibility and feeling guilty. Perceived as not being able to be supportive due to visiting restrictions. Restrictive contact with affected patient. Yearning for day-to-day status on patient. Trying to cope to meet patients' needs upon returning from treatment. Not feeling prepared or supported upon patient's return. |
| Forsberg et al., 2023 [27] | 14 family members of ICU COVID-19 patients. (Sweden) | Focus group (Qualitative) | Not being aware of the severity of the illness. Restricted visits, allowable only at the end of treatment and/or patient's life. Feeling of being unsupported and dealing with all issues by themselves. Indirect communication with ICU staff. Receiving information from ICU staff with little input from themselves. Dissemination of information to other family members. |
| Ghezeljeh et al., 2023 [28] | 12 family members of ICU COVID-19 patients. (Iran). | Interview (Qualitative) | Lack of knowledge of health status, lack of contact with ICU staff, insufficient information on treatments/medications and inadequate information to assist patients upon discharge. Lack of adequate treatment facilities. Neglect of families by ICU staff. Neglecting to take care of themselves. Feelings of being stigmatized. Anxiety, stress, fear of having a loved one in ICU. Witnessing the health struggles of loved ones. Feelings of perceived separation from loved ones. Blaming possible agents for causing illness. Anticipating worst-case scenarios (death). Responsibility of being a caregiver upon discharge. ICU staff are helpful in navigating health issues. Importance of social and financial supports. |
| Onrust et al., 2023 [29] | 56 COVID-19 patients and 67 family members of COVID-19 patients [a]. (Netherlands). | Interview (Qualitative) onrus Questionnaire/Survey (Quantitative) | Fear is associated with possible death of loved one. Hope for loved one's recovery. Lives contingent on loved one's treatment. Need for reassurance from ICU staff. Expressed need for peace and solitude. ICU staff are perceived as being helpful and caring to both patients and their families. Solace in expression (praying and lighting candles.) |

**Table 1.** *Cont.*

| Author(s) | Participants' Relationship to Victim; Country | Type of Study | Main Findings |
|---|---|---|---|
| Apostol-Nicodemus et al., (2022) [30] | 74 family members of COVID-19-infected patients. (Philippines) | Questionnaire/Survey (Quantitative) | 43% had anxiety within two weeks of patient's discharge, and 24% had anxiety within eight weeks of patient's discharge. 16% had depression within two weeks of patient's discharge, and 5% had depression within eight weeks of patient's discharge. 14% had mixed diagnoses (anxiety and depression) within two weeks of patient's discharge, and 4% had mixed diagnoses within eight weeks of patient's discharge. 10% had moderate family dysfunction within two weeks of patient's discharge, and 7% had moderate family dysfunction within eight weeks of patient's discharge. 10% had moderate family dysfunction within two weeks of patient's discharge, and 7% had moderate family dysfunction within eight weeks of patient's discharge. Decreased economic resource inadequacy within eight weeks of patient's discharge. At two weeks upon discharge, patient anxiety and inadequate family resources are linked to anxiety. At eight weeks upon discharge, patient anxiety and low educational attainment were linked to anxiety symptoms. At two weeks after discharge, patient depression was linked to depressive symptoms. |
| Galazzi et al., 2022 [31] | 56 family members of deceased ICU COVID-19 patients. (Italy) | Interview (Qualitative) Questionnaire/Survey (Quantitative) | 100% wished to see loved ones in ICUs; Only 11% did. 50% made a video call with a loved one. 96% of those who made a video call were content and would call again. 93% of those who did not perform a video call, regret not doing it. 45% could not hold funeral services by law. 38% used psychological support. Deceased patients' offspring are less likely to feel psychological distress than other familial relationships. |
| Greenberg et al., 2022 [32] | 62 surrogates of ICU COVID-19 patients. (U.S.) | Interview (Qualitative) | Difficulty in communication with ICU staff. Difficulty in communication with other family members. Difficulty in comprehending and tracking medical information. Distress with visiting restrictions in ICUs. |
| Heesakkers et al., 2022 [33] | 166 family members (Iran) | Questionnaire/Survey (Quantitative) | Symptoms of anxiety/depression are highest in the 3 months after loved ones' hospital discharges. Symptoms of anxiety/depression are higher in the 12 months after loved ones' discharges. |

**Table 1.** *Cont.*

| Author(s) | Participants' Relationship to Victim; Country | Type of Study | Main Findings |
|---|---|---|---|
| Khaleghparast et al., 2022 [34] | 324 family member participants (Iran) | Interview (Qualitative) Questionnaire/Survey (Quantitative) | Anxiety is linked to lack of knowledge of status of hospitalized loved one. Severe anxiety is higher in spouses of diagnosed loved ones. Anxiety was higher in lower-income individuals and females. |
| Khubchandani et al., 2022 [35] | 2797 Friends, family members, and acquaintances (US) | Questionnaire/Survey (Quantitative) | Knowing at least one family member or friend, being infected with COVID-19, or being hospitalized was more likely to increase the risk of anxiety/depression. |
| Jafari-Oori et al., 2022 [36] | 350 family caregivers of COVID-19 patients (Iran) | Questionnaire/Survey (Quantitative) | Younger than 20, married, having a child, employed in healthcare, COVID-free, not exercising, taking anti-anxiety medications, poor sleep, having another illness, higher income is associated with higher *Depression/Anxiety/Stress* scores (DAS) and increased *Fears associated with COVID-19* (FSV-19). Being married, being employed in healthcare, having COVID-19, not exercising, having another illness, having a higher income, and being younger were predictors of high anxiety scores. Being married, being employed in healthcare, not exercising, poor sleep, higher income, and being younger were predictors of high stress scores. |
| Mawaddah et al., (2022) [37] | 10 family members of COVID-19 patients. (Indonesia) | Interview (Qualitative) | Associated physical, economic, psychological, and social stigma burdens with caring for COVID-19-infected patients. Efforts to increase patient's immune system, family preventive efforts towards COVID-19 transmission, and trying to find out information on patient's health. Support from family and social support in caring for patient. Family adaptation to COVID-19 protocols, developing closer family relations, and spiritual improvement. Encountered obstacles at home in care of patient. Hoping of no social stigma with disease, the pandemic ends soon, and no other families are affected by disease. |

**Table 1.** *Cont.*

| Author(s) | Participants' Relationship to Victim; Country | Type of Study | Main Findings |
|---|---|---|---|
| Mejia et al., (2022) [38] | 3292 college students, 2789 [a] of them knew someone who died or were diagnosed with COVID-19 (Various countries in Latin America) | Questionnaire/Survey (Quantitative) | Anxiety is linked to friends dying, close relatives dying, and distant relatives dying. Severe/moderate depression is associated with age, class year, being from Honduras/Chile/Panama, a close relative dying, COVID-19 at home, and the respondent being ill. Severe/moderate anxiety is associated with gender, age, class year, being from certain countries, having a close relative, distant relative, or friend die, having a family member either ill at home or away from home, having a friend ill, or the respondent being ill. Severe/moderate stress is associated with being male, age, class year, having a romantic partner, being from Honduras/Chile/Panama, having a close relative dying, a distant relative dying, a friend dying, having an ill relative at home, a sick relative away from home, an ill friend, or respondent being ill |
| Nohesara et al., 2022 [39] | 12 family members of deceased COVID-19 patients. (Iran). | Interview (Qualitative) | Complex grieving processes with feelings of guilt and issues with emotional expression. New experiences associated with mourning. Developed more empathy for patients with COVID-19. Changing the meaning of death as a normal process of life. Increased need for support at work. |
| Robinson-Lane et al., (2022) [40] | 16 recently discharged ICU COVID-19 patients, and 16 family caregivers [a] (US) | Interview (Qualitative) | Taking on new responsibilities as caregivers. Managing mixed emotions (anxiety, grief, and joy) with their loved ones recently discharged from ICU. Engaging in preventive infection control against new infections. Trying to address patient independence with patients' perceived overbearing caregiving. The need for continued medical/emotional support in caregiving. |
| Rostami et al., 2022 [41] | 236 family caregivers of COVID-19 patients. (Iran) | Survey/Questionnaire (Quantitative) | 57% reported symptoms of depression. 70% reported symptoms of anxiety. 55% reported symptoms of stress. Female gender is associated with greater levels of stress. Self-employment is associated with greater levels of depression. |
| Bartoli et al., 2021 [42] | 14 family members of ICU COVID-19 patients. (Italy) | Interview/Questionnaire (Qualitative) | Fear is associated with the course of disease and the unknowns. Fear associated with prior knowledge of ICUs. Fear associated with information about the disease stems from the media. Feelings of trauma due to being away from loved ones due to restrictions of ICUs. Perceived having life on pause while waiting for news of a loved one from ICU staff. The realism that COVID-19 impacted their families. Feelings of guilt over a loved one's diagnosis. |

**Table 1.** *Cont.*

| Author(s) | Participants' Relationship to Victim; Country | Type of Study | Main Findings |
|---|---|---|---|
| Beck et al., 2021 [43] | 126 COVID-19 patients and 153 [a] family members (Switzerland) | Interview (Quantitative) | 16% had symptoms of anxiety, and 15% had symptoms of depression after 30 days discharge from hospital. Psychological distress associated with having children, not being employed, lower self-perceived overall health status, death of patient, use of psychotropic drugs, lower resilience, higher perceived stress, communicating through video calls or being able to visit the patient, higher perceived overall burden, increased worries about uncertain diagnosis and infection, higher burden of isolation measures and separation from patient, sport as coping strategy, relative was in contact with medical team, received information regarding prognosis, higher burden of not being able to visit patient, missing physical closeness, relative in quarantine, relationship with patient. |
| Borghi et al., 2021 [44] | 246 families of COVID-19 deceased victims. (Italy) | Interview (Qualitative) | Due to social distancing protocols, coming up with alternative ways to hold funerals. Rationalizing that most victims were older-that they would have eventually died from something else. Lockdowns provided time and space to process loved ones' deaths. Feeling helpless for not helping others and themselves. Conveying the news of the loved ones' deaths to others. |
| Chen et al., 2021 [45] | 10 family members of COVID-19 ICU patients. (US) | Interview (Qualitative) | Higher levels of stress or self-blame on a family member's diagnosis. Diagnosis associated with familial turmoil. Mixed views on video calls to ICU patients. Perceived poor closure to relationship after death. Needed information to treat and care for a loved one. Frustration is linked to a lack of access to knowledge of status. Appreciation of healthcare providers. |
| Jarial et al., 2021 [46] | 31 family members of COVID-19 patients treated in hospitals, 34 family members with patients treated at home, and 35 family members with no COVID-19 patients. (India) | Questionnaire/Survey (Quantitative) | Perceived stress was highest among family members who had a loved one treated in the hospital. |
| Joaquim et al., 2021 [47] | 9024 family members or friends of COVID-19 deceased victims. (Brazil) | Questionnaire/Survey (Quantitative) | Having friends or family members who have died is linked to more psychological distress. Already suffering from mental health issues (depression, anxiety, and psychotic) was associated with an exacerbation of symptoms from having lost a friend or family member. |

**Table 1.** *Cont.*

| Author(s) | Participants' Relationship to Victim; Country | Type of Study | Main Findings |
|---|---|---|---|
| Kentish-Barnes et al., 2021 [48] | 19 family members of deceased ICU COVID-19 patients. (France) | Interview (Qualitative) | Difficulty in communicating in-person and via telephone with ICU professionals. Communication problems with ICU professionals are due to the choice of words, pitch, or tone of conversations. Social distancing restrictions caused feelings of loneliness while loved ones were in the ICUs. Mixed emotions, with ICU professionals relaying both positive and negative news. Witnessing the care in ICUs made it feel personable. Meeting with ICU professionals established trust. Social distancing protocols made family members feel powerless. ICU professionals were viewed as the "go between" between loved ones and family members. Visits helped the family members feel that they were supporting, caring, and providing closure. Social distancing protocols led to both missing the final moments and/or modifications to funeral procedures, ceremonies, and mourning. |
| Koçak et al., 2021 [49] | 2047 had someone who had been ill or died from COVID, and 1240 did not. (Turkey) | Questionnaire/Survey (Quantitative) | Anxiety, stress, and depression were higher in those who had friends or loved ones who had been diagnosed with or died from COVID-19. Fear of COVID-19 is associated with anxiety, stress, and depression |
| Nakhae et al., 2021 [50] | 16 family caregivers of COVID-19-infected patients. (Iran) | Interview (Qualitative) | Fear of death of a loved one and the ambiguous nature of the disease. Caring difficulties and trying to provide best care. Quarantine issues (isolation) and social support. Technology (internet-based) was both harmful and helpful. Distrust of hospital care; preference for caring at home. Home care led to long-lasting positive experiences. |
| Orsini et al., 2021 [51] | 58 parents of COVID-19-infected children and 39 parents of non-COVID-infected children. (Italy) | Questionnaire/Survey (Quantitative) | Having children who tested COVID-19 positive, suffering economic hardships, being quarantined, or having a close relative who was diagnosed with COVID-19 were more likely to report moderate/severe anxiety. Having children who tested positive for COVID-19, suffering from economic hardships, and being quarantined were more likely to report moderate/severe depression. |

**Table 1.** *Cont.*

| Author(s) | Participants' Relationship to Victim; Country | Type of Study | Main Findings |
|---|---|---|---|
| Picardi et al., 2021 [52] | Eight participants, all but one were family members of COVID-19 hospitalized patients. (Italy) | Focus Group (Qualitative) | Need to be constantly informed of the treatments/progress. Perceived as being non-effective or impotent. Reported difficulties with communication (phone, internet) with healthcare providers. Shared their day-to-day burdens (economic issues, lack of medical access) Psychological issues of having a hospitalized family member (anxiety) |
| Prakash et al., 2021 [53] | 93 patients who had COVID-19 and 54 family members (India) | Questionnaire/Survey (Quantitative) | 17 (25%) of infected individuals reported some form of depression (mild, moderate, severe, or extremely severe) vs. 22 (41%) of family members of infected individuals. 21 (31%) of infected individuals reported some form of anxiety (mild, moderate, or severe) vs. 25 (46%) of family members of infected individuals. 9 (13%) of COVID-19-infected individuals reported some form of stress (mild, moderate, severe, or extremely severe) vs. 16 (30%) of family members of infected individuals |
| Rahimi et al., 2021 [54] | 13 family caregivers of COVID-19-infected patients. (Iran) | Interview (Qualitative) | Perceived difference in care for COVID-19 vs. other diseases. The unexpected reoccurring symptoms. Needs of caregivers are not met. Need information to treat/care for a loved one. Lack of access to healthcare services. Financial problems with caretaking. Unpleasant social, physical, and psychological experiences with care Spirituality/social support strengthen resolves. Coping methods are used for stress. Caregiving led to some positive experiences and a sense of self-growth. |
| Selman et al., 2021 [55] | Twitter data was gathered. 196 tweets from 192 friends and family members of deceased COVID-19 patients. (Various countries) | Tweets-Technology Qualitative Study | Tweeted about social restrictions limiting ability to visit places of care. Tweets about family members dying alone or without a proper farewell. Tweets about the emotional impacts of having a family member severely affected by COVID-19 (e.g., government response to virus, perceived public apathy). Tweets about a lack of social support after the death of loved ones or funerals not held as the victim would have wished. Tweets about the importance and support of PPE usage, social distancing, and hygienic practices. |

**Table 1.** *Cont.*

| Author(s) | Participants' Relationship to Victim; Country | Type of Study | Main Findings |
|---|---|---|---|
| van Veenendaal et al., 2021 [56] | 50 COVID-19 patients and 67 family members of COVID-19 patients [a] (Netherlands) | Questionnaire/Survey (Quantitative) | Showed good physical functioning. 64% went back to work after six months of ICU discharge. Showed good psychological functioning after three and six months after victim's ICU discharge. 63% reported impaired well-being from the mandatory physical distance from the victim while in the ICU. |
| Xu et al., 2021 [57] | 1274 non-COVID social contacts and 173 social contacts of COVID-19 patients [a] (China). | Questionnaire/Survey | Social contacts of COVID-19 patients were more likely to have anxiety symptoms, depressive symptoms, suicidal ideations/thoughts, PTSD symptoms, somatic symptoms, poorer meaning in life, loneliness, lower HRQOL, more COVID-19-related symptoms, and lower satisfaction with life. Social contacts of COVID-19 patients were more likely to have different perceptions about the epidemic (more worries about infection), a higher perceived risk of being infected, a longer perceived time for successful epidemic control, pay more to prevent infections, have visits to a doctor in the past four weeks, and report lower perceived self-efficacy. |
| Zhao et al., 2021 [58] | 1290 had no close contact with COVID-19 patients, and 1169 had close contact with a COVID-19 patient [a]. (China) | Questionnaire/Survey (Quantitative) | Close contact participants were more likely to report severe depression and fatigue. Close contact participants were more likely to suffer from depression and fatigue if they were younger, had economic issues due to the pandemic, or had a perception of poor or fair health. Close contact participants were likely to suffer from fatigue because they frequently used mass media. |
| Mirzaei et al., 2020 [59] | 210 family caregivers of both inpatient and outpatient COVID-19 patients. (Iran) | Questionnaire/Survey (Quantitative) | Male caregivers are more likely to suffer from objective, subjective, or objective–subjective burdens. No difference in the mean scores of the total caregiver burden by gender. |
| Mohammadi et al., 2020 [60] | 16 family members of deceased COVID-19 victims. (Iran) | Interviews (Qualitative) | Intense emotional shock from losing a loved one. Perceived guilt and fear of transmitting disease to a loved one. Lack of proper closure due to sudden death. Lack of proper burial or unreligious burial due to social restrictions Fear of the future due to family instability caused by the death of a loved one. Feeling of stigmatization by society due to a loved one dying. |

**Table 1.** *Cont.*

| Author(s) | Participants' Relationship to Victim; Country | Type of Study | Main Findings |
|---|---|---|---|
| Rizvi Jafree et al. [61] | 20 family members of hospitalized COVID-19 patients. (Pakistan) | Interviews (Qualitative) | Social stigmas: Police intimidation, maltreatment by hospital staff, false test results used as revenge, blame/rejection by others, physicians facing stigma from other physicians, discrimination at work, shift locality, and difficulty commuting for necessities due to maltreatment. Struggles: Social distancing, inadequate knowledge of disease, having to replace mothers for care of children, depression and sleeping problems, troubles adjusting to the post-COVID world, and anxiety for children. Strengths: Praying and patience, spirituality, support and assistance from daughters, thankful for what one has, doing household activities one has never conducted before, planning a healthy future, using the media as a source of awareness and learning, and exercise. |
| Tanoue et al., 2020 [62] | 16,402 participants [b] (Japan) | Questionnaire/Survey (Quantitative) | Higher rates of stress are associated with family members being diagnosed. |

[a] = Only used the information associated with those familiar with COVID-19 victims/patients. [b] = No breakdown of participants.

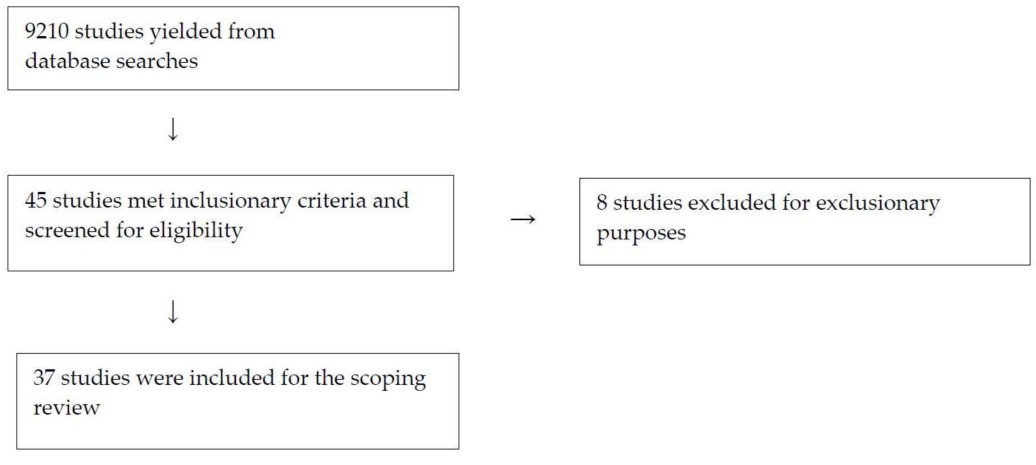

**Figure 1.** PRISMA Flow of Studies' Selection.

Descriptive headings were included for each study in Table 1: Author(s), Participants Relationship to Victim(s), Country; Type of Study; Main Findings. The results from the quantitative studies were included if they met statistical significance (i.e., $p < 0.05$). Owing to the prevalence of results yielded from the searches, the findings were placed into corresponding, combined categories (Table 2). This review only included categories that yielded similar findings in three or more studies. Next, the findings were amalgamated into 16 broad categories.

**Table 2.** Grouped Categories from the Findings.

| Category | Studies |
|---|---|
| Psychosocial feelings: Anxiety, stress, distress, and fear associated with COVID-19-infected loved one. | Ghezeljeh et al., 2023 [28]; Apostol-Nicodemus et al, 2022 [30]; Galazzi et al., 2022 [31]; Heesakkers et al., 2022 [33]; Khaleghparast et al., 2022 [34].; Khubchandani et al., 2022 [35]; Jafari-Oori et al., 2022 [36]; Mejia et al., 2022 [38]; Robinson-Lane et al., 2022 [40]; Rostami et al., 2022 [41]; Beck et al., 2021 [43]; Chen et al., 2021 [45]; Jarial et al., 2021 [46]; Joaquim et al., 2021 [47]; Koçak et al., 2021 [49]; Orsini et al., 2021 [51]; Picardi et al., 2021 [52]; Prakash et al., 2021 [53]; Rahimi et al., 2021 [54]; van Veenendaal et al., 2021 [56]; Xu et al., 2021 [57]; Zhao et al., 2021 [58]; Rizvi Jafree et al., 2020 [61]; Tanoue et al., 2020 [62]. |
| Perceived Issues with healthcare system(s): Wanting more information on a loved one or how to properly care for them upon discharge. Lack of clear communication. Lack of follow-up care upon returns. Not knowing severity of illness. | Berntzen et al., 2023 [26]; Forsberg et al., 2023 [27]; Ghezeljeh et al., 2023 [28]; Onrust et al., 2023 [29]; Greenberg et al., 2022 [32]; Robinson-Lane et al., 2022 [40]; Bartoli et al., 2022 [42].; Chen et al., 2021 [45]; Kentish-Barnes et al., 2021 [48]; Nakhae et al., 2021 [50].; Picardi et al., 2021 [52]; Rahimi et al., 2021 [54]; Rizvi Jafree et al., 2020 [61]. |
| Importance of support: Social, emotional, and financial support in having a COVID-19 loved one or taking care of one. | Berntzen et al., 2023 [26]; Forsberg et al., 2023 [27]; Ghezeljeh et al., 2023 [28]; Apostol-Nicodemus et al., 2022 [30]; Galazzi et al., 2022 [31]; Mawaddah et al., 2022 [37]; Nohesara et al., 2022 [39]; Robinson-Lane et al., 2022 [40]; Picardi et al., 2021 [52]; Rahimi et al., 2021 [54].; Selman et al., 2021 [55]. |
| Responsibility/caring for COVID-19 patients: Being responsible for taking care of a COVID-19 patient or overseeing the family due to the patient's absence. Feeling unprepared. | Berntzen et al., 2023 [26]; Forsberg et al., 2023 [27]; Ghezeljeh et al., 2023 [28]; Mawaddah et al., 2022 [37]; Robinson-Lane et al., 2022 [40]; Chen et al., 2021 [45]; Nakhae et al., 2021 [50]; Picardi et al., 2021 [52]; Rahimi et al., 2021 [54]; Mirzaei et al., 2020 [59]; Rizvi Jafree et al., 2020 [61]. |
| Visiting restrictions and isolation: Issues with restrictions on visiting patients in healthcare facilities. Feelings of isolation from an infected loved one and society. | Berntzen et al., 2023 [26]; Forsberg et al., 2023 [27]; Ghezeljeh et al., 2023 [28]; Galazzi et al., 2022 [31]; Greenberg et al., 2022 [32]; Kentish-Barnes et al. 2021 [48]; Nakhae et al., 2021 [50]; Selman et al., 2021 [55]. |
| Guilt feelings or emotional impact: Feelings of guilt or emotional distress from having a loved one with COVID-19. | Berntzen et al., 2023 [26]; Nohesara et al., 2022 [39]; Robinson-Lane et al., 2022 [40]; Bartoli et al., 2021 [42]; Chen et al., 2021 [45]; Selman et al., 2021 [55]; Mohammadi et al., 2020 [60]. |
| Funeral services or mourning restrictions: Due to COVID-19 protocols, funeral or other mourning practices were modified or restricted. | Galazzi et al [31]., 2022; Nohesara et al., 2022 [39]; Borghi et al., 2021 [44]; Kentish-Barnes et al., 2021 [48]; Mohammadi et al., 2020 [60]. |
| Needing peace and solitude: Alone time. Meditation time. Spirituality. | Onrust et al., 2023 [29]; Mawaddah et al., 2022 [37]; Borghi et al., 2021 [44]; Rahimi et al., 2021 [54]; Rizvi Jafree et al., 2020 [61]. |
| Worrying about self or others/Neglect: Worrying over family health, neglecting care for self or others. | Berntzen et al., 2023 [26]; Ghezeljeh et al., 2023 [28]; Robinson-Lane et al., 2022 [40]; Borghi et al., 2021 [44]; Rahimi et al., 2021 [54]. |
| Feelings of stigma: Being or feeling stigmatized for having a COVID-19-infected patient. | Ghezeljeh et al., 2023 [28]; Mawaddah et al., 2022 [37]; Mohammadi et al., 2020 [60]; Rizvi Jafree et al., 2020 [61]. |
| Category | Studies |
| Healthcare system being helpful: The healthcare system was perceived to have assisted in several ways. | Ghezeljeh et al., 2023 [28]; Onrust et al., 2023 [29]; Chen et al., 2021 [45]; Kentish-Barnes et al., 2021 [48]. |
| Media: How the media (e.g., social, print, TV) is either perceived as helpful or harmful in covering or reporting information on COVID-19. | Robinson-Lane et al., 2022 [40]; Bartoli et al., 2021 [42]; Nakhae et al., 2021 [50]; Rizvi Jafree et al., 2020 [61] |

**Table 2.** *Cont.*

| Category | Studies |
|---|---|
| Being the point of contact: Having to contact other members of family about patient's health or having difficulty contacting them. | Forsberg et al., 2023 [27]; Galazzi et al., 2022 [31]; Borghi et al., 2021 [44]. |
| Fear of death or grief over losing a COVID-19-infected loved one. | Onrust et al., 2023 [29]; Nakhae et al., 2021 [50]; Mohammadi et al., 2020 [60]. |
| Perceived inadequate closure at end of life: Not having the proper farewell with victim. | Chen et al., 2021 [45]; Selman et al., 2021 [55]; Mohammadi et al., 2020 [60]. |
| PPE and other safety measures: The Importance of using personal protective equipment and social distancing. | Mawaddahi et al., 2022 [37]; Robinson-Lane et al., 2022 [40]; Selman et al., 2021 [55]. |

The three most prevalent findings were: Psychosocial feelings (24 studies); Perceived issues with the healthcare system(s) (13 studies); Importance of support (11 studies); and Responsibility/caring for COVID-19 patients (11 studies) (Table 2). The rest of the findings are displayed at reduced frequencies. Both quantitative and qualitative studies comprised 17 of the types of studies (Table 3). Three of the studies utilized a mixed framework, both quantitative and qualitative. All quantitative studies utilized questionnaire/survey methods. All but three qualitative studies employed the interview method. Iran was the country that yielded the most studies (10). Assessing the study population used in the studies, those that contained 0–100 participants were the most prevalent (19).

**Table 3.** Descriptive Data of the Studies.

| Types of Study |
|---|
| Quantitative = 17 Studies<br>Qualitative = 17 studies<br>Mixed = 3 studies |
| Most Common Type of Quantitative Study |
| Questionnaire/Survey = 17 |
| Most Common Type of Qualitative Study |
| Interview = 14 |
| The Top Three Countries with the Most Studies |
| Iran = 10 |
| Italy = 5 |
| United States = 4 |
| Number of Participants in the Study Populations |
| 1–100 = 19 |
| 101–1000 = 11 |
| 1001+ = 7 |

## 4. Discussion Section

Psychological states (e.g., anxiety, stress, depression) linked to COVID-19 diagnoses in relatives or close friends were the most prevalent findings from the review. This finding comprised more than half of all selected studies. This was not surprising due to the range of emotions that may arise from having a friend or family member either diagnosed with or having died from the virus. Coupled with the fact that caregiving for severely ill patients outside of the healthcare setting is mostly provided by relatives and close friends [64]. This finding is consistent with other studies assessing informal caregiving for other severe illnesses [65,66] The death of a relative or close friend may cause emotional anguish for

some survivors. The loss of a family member may cause not only mental distress for survivors but also the exacerbation of current mental illnesses or disorders [48].

Caregiving, conducted for long periods of time, may increase the risk of developing psychological or emotional issues. The severity and duration of psychological disorders in affected patients are correlated with increased risks of anxiety and depression in caregivers [67]. Some personal characteristics of caregivers may be linked to greater susceptibility to psychological issues. In one of the selected studies from the review, personal characteristics such as being married, having a higher income, and not exercising were positively associated with higher risks for developing depression, anxiety, and stress in caregivers [37]. The results taken from the review were comparable to those derived from similar studies looking into the effects of informal caregiving, regardless of the illnesses, during the pandemic's peak. Caregivers reported higher levels of depression, anxiety, fatigue, sleep issues, and food insecurity when compared to non-caregivers [68]. Some psychological states, like loneliness, may increase the risk of developing symptoms associated with depression [69].

Perceived problems with the healthcare system(s) were a common theme throughout the review. Many of these types of problems stem from miscommunication between healthcare professionals and caregivers pertaining to the status of a hospitalized patient. Several studies from the scoping review reported that family members lamented difficulties in both communication and comprehension of the medical prognoses of patients [49,53]. Some family members expounded on the importance of being informed of the hospitalized patient's progress throughout treatment [53]. Access to staff and being informed of a patient's status were cited as being effective in bolstering the lines of communication and understanding between healthcare staff and relatives [70].

Healthcare workers may use verbiage or assume that caregivers understand directives. Healthcare workers may need to be cognizant of the fact that many caregivers may have low health literacy, which substantiates their need for assistance in understanding health information. Therefore, clarification or probing for questions can be important. Caregivers should be informed alongside the patient, be it through verbal or written instruction [71]. Ideally, caregivers should be trained by healthcare providers to effectively monitor current symptoms and report any noticeable exacerbation of symptoms [72].

During a pandemic, family members, especially caregivers, may feel despondent or overwhelmed with day-to-day responsibilities. They may feel alone or not adequately supported when the COVID-19-infected patient returns from hospitalization or treatment [27,28]. Caregiving for severe or chronic diseases may lead to increased levels of stress [73] or depression [74]. Greater family support is linked to a reported greater health-related quality of life in caregivers [75]. Financial woes or economic pressures, which personified COVID's peak period, may also lead to perceived burdens for informal caregivers [76].

While COVID-19 may not be as time-consuming in caregiving or treatment as compared with other more chronic or severe diseases, support is still recommended for caregivers. Both tangible and informational support are integral in assisting caregivers of patients with severe or chronic illnesses [77]. In one study, caregivers cited financial assistance as a crucial source of support during trying times [78]. Providing ample social support can be the impetus for alleviating perceived burdens among caregivers [79]. Though some social support systems may only have slight improvements in health or may lead to negative effects on sufferers [80]. One review study assessing self-management programs for asthma sufferers found that some participants reported a lack of empathy or reluctance from their loved ones to accept the severity of their illness [81].

During times of medical emergencies, family members and close friends may find themselves responsible for the care of an infected loved one [41]. Some caregivers report an array of different burdens (e.g., economic, physical, and psychological) in the long-term care of others [38]. Devoting more time to caregiving was positively associated with poorer mental health outcomes [82]. Females are more likely to devote more time

to caring for infected family members, and as such, they are more likely than males to suffer from higher perceived burdens [83]. Though not all experiences of caregiving are reportedly negative, some caregivers report feeling a sense of positive accomplishments and self-growth throughout their experiences [55].

During the peak of communicable disease control, social distancing protocols are rudimentary precautions to reduce the spread of transmission and protect immunocompromised individuals. Though many family members try to comply with the protocols [51], the restrictions may still have negative impacts on both the patients and their family members [49,62]. These needed protections may lead to a greater likelihood of adverse outcomes. A literature review studying the effects of hospital restrictions on visitations found that family members of COVID-19-infected patients were more likely to suffer from anxiety, excessive worrying, and even increased disturbances at home [84]. Being separated from infected loved ones may add to feelings of foreboding or dread among relatives and friends [56]. Some may anticipate the worsening of the hospitalized individual's health stemming from social isolation. McCleary et al. examining COVID-19-era long-term care facilities during the severe acute respiratory syndrome (SARS), reported that relatives perceived drastic declines in the physical and emotional health of their loved ones due to visiting restrictions.

Being isolated from loved ones may be traumatic for both hospitalized patients and their family members. It may be ideal to include more personal contact times between patients and their relatives, regardless of medical emergencies. Allowing for more personal contact time, regardless of the restrictions, may help mitigate the maladaptive effects associated with feelings of isolation [85]. Some hospitals have found ways to connect families to their hospitalized loved ones via implemented technological solutions (virtual visits) [86]. A recent study found that while telehealth or other forms of technological communication are preferred to no means of communication, face-to-face visits are still considered ideal by most families [87].

## 5. Limitations to the Study

This scoping review only used those studies that were selected from the database searches. Therefore, some studies contained in other databases may be missing from this review. Some of these potentially missing studies may have yielded results contrary to the cumulative findings from the present review. The 37 published studies may serve as a relatively small sample. A larger sample size may have also yielded a different set of results. The authors included only those search words, or MeSH terms, that they believed would generate the most relevant results. Excluding non-English language published studies may have precluded some prospective studies from being used in the review. It could be surmised that they may have been able to attain additional studies through the utilization of other search words or MeSH terms. The selected studies did not indicate from which COVID wave the results were derived. The difference in severity of health issues stemming from the different waves may impact how COVID infection is perceived by loved ones of infected individuals. Most of the studies did not report being either longitudinal or cross-sectional in design. Temporality may lead to changes in opinions or perspectives among participants. Lastly, most of the selected studies included self-reported data from participants, which may lead to potential biases that may affect the results published from those studies.

## 6. Conclusions

This scoping review may provide the impetus for future interventions targeting the alleviation of the psychological or emotional issues associated with losing close friends or relatives to severe diseases. The results from the scoping review found that psychosocial feelings (e.g., anxiety, stress, and fear) were the most prevalent of the grouped categorical findings. This can be expected due to the possible emotional toll that infection may take on a victim's loved ones. Also, the findings from this study could be ideally utilized as a

resource for those researchers studying the effects of future pandemics on the psychological stressors or daily hassles of non-professional caregivers, primarily those who are relatives or friends of the affected person.

Usually, implemented interventions target the mental health needs of the victim. As such, interventions need to target alleviating the emotional or psychological states that may afflict the families or friends of a victim. Family members and close friends increasingly serve as the primary caregivers for loved ones who suffer from severe diseases. Therefore, mental health services (e.g., counseling, medications) may assist the relatives of severely diseased individuals in coping with the daily stressors of life.

**Author Contributions:** Conceptualization, M.W.; Methodology, M.W., T.C., K.E.; Software, M.W., T.C., K.E.; Validation, M.W. Formal Analysis, M.W.; Investigation, M.W., T.C., K.E.; Resources, N/A; Data curation, M.W.; Writing—original draft preparation, M.W., T.C., K.E. Writing—review and editing, M.W.; Visualization, M.W.; Supervision, N/A.; Project administration, N/A; Funding. All authors have read and agreed to the published version of the manuscript.

**Funding:** This research received no external funding.

**Institutional Review Board Statement:** Not applicable.

**Informed Consent Statement:** Not applicable.

**Data Availability Statement:** We have no new available data to share.

**Conflicts of Interest:** The authors declare no conflict of interest.

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
