# Peer review of "The Extent and Impact of COVID-19 Infection among Family and Friends: A Scoping Review"

_2673-8430, doi:10.3390/biomed3030028_

Round 1

Reviewer 1 Report

Dear Authors,

The topic is interesting and relevant. I concur with your rationale that most studies focus more on the patients than the families.

The following are my comments and suggestions:

1.  You claimed that your study is the first scoping review related to your identified concept/focus, however you also mentioned in your limitations regarding other databases you did not use in your study which means that you have not exhausted all evidence that your study is the first scoping review done. I suggest that you add more specific modifiers to this claim.

2. Line 45- Be consistent with the use of COVID-19 and just COVID

3. Line 119-121- I found just one aim and one research question. Kindly revisit your statement.

4. Methods- I found this section very limited. I suggest to reorganize this section to be helpful to other researchers and readers who are interested in scoping reviews.

Specify subsections such as: Preliminary search, Inclusion criteria, Search Strategy, Source of evidence selection, Data extraction, and Analysis of evidence

In data analysis, discuss how you came up with your themes/categories.

5. Add in your discussion and conclusion on the scope of studies done related to research designs, instruments, countries, population...

6. In your conclusion, add a very brief summary answer to your research aim/research question before your recommendations.

Thank you.

Few grammatical errors exist.

Reviewer 2 Report

the authors analyzed several articles that dealt with reactions in the population in the phases of covid-19.
in particular the authors highlighted the effect of the support of family and friends or lack thereof.
Some suggestions are proposed

The introduction is divided into several paragraphs which are very short and one could think of reorganizing the introduction section with fewer divisions.

Articles published in different years were used: from 2020 to 2023. As is known, the reactions of the population varied according to the different waves of Covid-19.
Not only the restriction measures have had an impact but also other factors, such as knowledge of the virus and its transmission, the development of treatment protocols, the severity of the virus infection which has decreased over time from the Alpha type to Omicron , which was significantly less offensive.
It is considered important that the Authors indicate how many studies refer to the first, second or third wave of Covid-19. Being a qualitative study, the Authors could explain how the results have changed according to the Covid-19 waves.
For example, the studies by Simione et al. 2021; 2022 highlighted how Covid-19 was closely associated with death anxiety during the first Covid-19 wave.
Death anguish decreased in the second wave when vaccines were available

The Authors do not seem to emphasize results in contrast with their hypothesis (null hypothesis) which should be considered in the context of scientific research, even fine qualitative studies. Not only has proximity and social or family support reduced the severity of symptoms in the population, but also the perception of the presence of more effective interventions and remedies, such as vaccines.
Furthermore, the Authors do not seem to have highlighted studies in which social and family support did not show any effect in reducing the stress perceived by the population. For example, the study by Vagni et al. 2022, where family and social support showed no predictive effect on the reduction of stress manifestations    

The English text appears clear

Round 2

Reviewer 2 Report

I carefully reviewed the manuscript. I think however that the authors did not respond to the requested revisions. if the authors are unable to outline what is required in point 2, this represents an important limitation of their study which should at least be highlighted in the limitation section. instead it would have been more correct to re-examine the material also on the basis of the fact that the studies are different: some longitudinal, others not. would be information that the authors should indicate

the requested revision and the response of the authors on the point are reported

Reviewer #2’s Comment #2: As is known, the reactions of the population varied according to
the different waves of the Covid-19.
Not only on the restriction measures have had an impact but also other factors, such as
knowledge of the virus and its transmission, the development of treatment protocols, the severity
of the virus infection which has decreased over time from the Alpha type to Omicron , which was
significantly less offensive.
It is considered important that the Authors indicate how many studies refer to the first, second or
third wave of covid-19. Being a qualitative study, the Authors could explain how the results have
changed according to the Covid-19 waves.
For example, the studies by Simione et al. 2021; 2022 highlighted how Covid-19 was closely
associated with death anxiety during the first Covid-19 wave.
Death anguish decreased in the second wave when vaccines were available.
Our Response to Reviewer #2’s Comment #2: Thank you for the insightful comment.
Unfortunately, this cannot be added to our present study for several reasons. 1) There is no
discernable method to break down the studies from our scoping review by COVID-19 wave.
Rarely did the selected studies mention the COVID-19 wave. 2) Some of the studies were
longitudinal in length, there is no clear indicators to which wave(s) was being studied.
We do agree with the reviewer that the different waves may have an impact on perceived severity
of the disease by an infected individual’s loved ones. This could lead to further studies building
from a hypothesis associated with perceived severity disease contingent on the waves.

the additional revision point and the authors' response are reported

Reviewer #2’s Comment #3: The authors do not seem to emphasize results in contrast with their
hypothesis (null hypothesis) which should be considered in the context of scientific research,
even fine qualitative studies. Not only has proximity and social or family support reduced the
severity of symptoms in the population, but also the perception of the presence of more effective
interventions and remedies, such vaccines.
Furthermore, the Authors do not seem to have highlighted studies in which social and family
support did not show any effect in reducing stress perceived by the population. For example, the
study by Vagni et al. 2022, where family and social support showed no predictive effect on the
reduction of stress manifestations.
Our Response to Reviewer #3’s Comment #2: We appreciate the suggestions here. Please be
cognizant that this is a scoping review, usually scoping reviews do not test research/null
hypotheses. We have reverted back to answering the research question from a suggestion from
the other reviewer. Yes, we agree that the knowledge of the effectiveness of vaccinations may
have impact on how loved ones view the severity of a transmission of a disease. But to our
surprise, it was barely mentioned in the articles we collected as impacting the loved ones of
infected individuals.

One cannot agree on the authors' response. the literature is immense on the subject of covid-19 and having analyzed only 37 articles is a major limitation (already indicated by the authors) which should be better explained. It seems unusual that previous research has not linked family and friend support along with other variables of actual efficacy with respect to Covid-19. if so, it represents a further limit to the type of investigation that the authors have carried out. moreover, a study has been indicated which is in contrast with their conclusions. being a review of studies, studies showing contrasting results should also be indicated. if what was stated by the authors were true, it would represent a limit to the validity of the study as only studies confirming the importance of the support would have been selected, while in reality there are studies that have not found confirmation of the effectiveness of the support

Author Response

Thank you for your suggestions. We may have taken into consideration your suggestions and have included most of them into the revisions.

Round 3

Reviewer 2 Report

after the review process the manuscript can be published